# Alginate-Based 3D A549 Cell Culture Model to Study *Paracoccidioides* Infection

**DOI:** 10.3390/jof9060634

**Published:** 2023-05-31

**Authors:** Kelvin Sousa dos Santos, Lariane Teodoro Oliveira, Marina de Lima Fontes, Ketylin Fernanda Migliato, Ana Marisa Fusco-Almeida, Maria José Soares Mendes Giannini, Andrei Moroz

**Affiliations:** 1Department of Clinical Analyses, School of Pharmaceutical Sciences, São Paulo State University (UNESP), Araraquara 85040-167, São Paulo, Brazil; 2Centro Universitário Central Paulista—UNICEP, São Carlos 13563-470, São Paulo, Brazil

**Keywords:** 3D culture, scaffolds, sodium alginate, fungi infection

## Abstract

A three-dimensional (3D) lung aggregate model based on sodium alginate scaffolds was developed to study the interactions between *Paracoccidioides brasiliensis* (Pb) and lung epithelial cells. The suitability of the 3D aggregate as an infection model was examined using cell viability (cytotoxicity), metabolic activity, and proliferation assays. Several studies exemplify the similarity between 3D cell cultures and living organisms, which can generate complementary data due to the greater complexity observed in these designed models, compared to 2D cell cultures. A 3D cell culture system of human A549 lung cell line plus sodium alginate was used to create the scaffolds that were infected with Pb18. Our results showed low cytotoxicity, evidence of increased cell density (indicative of cell proliferation), and the maintenance of cell viability for seven days. The confocal analysis revealed viable yeast within the 3D scaffold, as demonstrated in the solid BHI Agar medium cultivation. Moreover, when ECM proteins were added to the alginate scaffolds, the number of retrieved fungi was significantly higher. Our results highlight that this 3D model may be promising for in vitro studies of host–pathogen interactions.

## 1. Lay Summary

This paper reports a new manner/technique to cultivate human cells and study their characteristics and behavior when close to microorganisms, such as fungi, inside a 3D structure. The findings revealed that this is an efficient technique to study human cells: they remain alive and proliferate.

## 2. Introduction

Current research in fungi–host interaction in vitro relies mainly on two-dimensional (2D) single-cell cultures. Although these methods have contributed to gathering knowledge about these interactions, infectious disease mechanisms, and the development of new therapeutic agents, they lack many essential features that are present in vivo and/or in animal models [1,2,3]. Furthermore, there is a lack of microenvironmental data regarding the respiratory tract’s main entry point for many fungi [4,5,6]. Three-dimensional (3D) bioprinted models can help to simulate this microenvironment, and the recent bioprinting of live human cells demonstrated that effective in vitro replication is achievable. These advances in 3D bioprinting opened a new frontier in research and advertised an era of more advanced therapeutics in mycology due to this more robust model platform, when compared to 2D cell cultures, for fungi–host interaction studies [7].

There are several in vitro assay methods that are based on 2D cell cultures; these approaches lead to new therapies and allow the elucidation of mechanisms involved in the course of diseases. Human cells have been grown as two-dimensional (2D) monolayers on flat plastic surfaces for decades and are still used in many research guides due to their easy handling and high performance, which yields faster results and low maintenance costs [8]. Two-dimensional cell cultures have provided important insights into drug screening, cellular metabolism, toxicity evaluation, and host–pathogen interactions before in vivo steps [9,10]. However, these 2D cultures lack a more robust tissue architecture, which can mimic the tissue complexity a little more, with the expression of several proteins that are involved in pathogenicity mechanisms.

Performing a 2D co-culture, using cells with different phenotypes, increases the similarity regarding the cellular heterogeneity observed in vivo models; however, this is still not enough when compared to living tissues [11,12,13,14,15,16]. In this sense, a 3D co-culture is a more promising strategy for creating integrated cross-communication networks in vitro [17]. There are many attributes that make them resemble healthy tissues, such as extracellular matrix (ECM) deposition, biomechanical forces, higher cellular communication through their surface receptors, and metabolic agents [11,13,16,17,18]. As they were explored, these 3D cultures were significantly improved with the recent advent of decellularized scaffolds, bioactive scaffolds, organoids, and organ-on-a-chip, which are used in different applications in biomedical/biotechnological research [8].

Sophisticated 3D cultures using scaffolds—or scaffolding—can be generated using several natural polymers (collagen, elastin, fibrin, chitosan, alginate, fibrinogen, and platelet-rich plasma), synthetic materials, which include PLLA (poly (L-lactic acid)), PGA (poly (glycolic acid)), elastomeric polyurethanes, decellularized tissues and organs, and others [18,19,20,21,22,23]. The main challenge in scaffold designing is to mimic the presence of the ECM, a network of fibrous proteins, primarily collagen, embedded in a highly hydrated gel of glycosaminoglycans, proteoglycans, and glycoproteins [11,24,25]. The cell behavior in these systems can be influenced by the scaffold structure or matrix and cell type. Moreover, the material of choice depends markedly on its intended application [17,26,27,28].

Probably the most common sodium-alginate-based scaffolds can be generated in hydrogels, films, and nanofibers [25]. The properties observed in the alginate hydrogel, such as swelling, porosity, and gel resistance, result from gel formation kinetics, which involves the chemical structure, the size of the alginate molecule, and the ion exchange conditions [29,30,31,32,33,34,35,36,37]. This approach has a low cost, low toxicity, and biocompatibility. Furthermore, its porosity allows the diffusion of nutrients from the culture medium to the cells to induce the synthesis of different components (mainly the ECM) and, depending on the size of the pores, it promotes cell migration [31]. Another interesting point is that the ionic crosslinking process is instantaneous and reversible, which can be undone to recover the encapsulated cells. In addition, the alginate scaffold can be assembled in different geometrical forms and approaches, such as beads, sheets, and layers [25,38,39,40].

Previously, it was reported that HepaRG cells self-assembled in alginate beads exhibited the ability to form cellular aggregates that remained highly viable for a long time. Simultaneously, cells showed a fully differentiated phenotype with decreased hepatoblast markers and an increased level of mature hepatocyte markers. In addition, hepatocytes microencapsulated in alginate beads revealed solid metabolic activity and the ability to eliminate toxins, including lactate and ammonia. The authors concluded that this 3D system maintains differentiated cells and could be used as a human bioartificial liver in extracorporeal treatments for patients with fulminant liver failure [14,41].

Even though 3D scaffolds during in vitro studies have been developed for bacteria and virus infection, few reports explored fungal infection studies [1,2,3]. Most lung models used so far are statically cultured and therefore are not subjected to shear stress. Furthermore, these models rarely consider the impact of additional microbial community members, such as the lung microbiota, on the infection process [3].

*Paracoccidioides* spp. is a thermally dimorphic fungus responsible for causing paracoccidioidomycosis (PCM), and *P. brasiliensis* is a common species, which, according to information from the Ministry of Health, is the eighth most common cause of mortality among parasitic infectious diseases in Brazil. In recent work surveying autopsies between 1930 and 2015, paracoccidioidomycosis (24%) was the most frequent infection [42]. Various approaches have been developed to study *Paracoccidioides*–host interaction [3,5,6,43,44,45,46,47]. However, according to the literature, we still do not have studies on the interaction of this fungus in 3D models.

This work proposes the development of an alternative 3D model based on sodium alginate scaffolds to evaluate host–pathogen interactions between human lung cells and *Paracoccidioides*, which causes prevalent systemic mycosis in Latin American countries.

## 3. Material and Methods

### 3.1. Two-Dimensional Cell Culture

The immortalized human lung adenocarcinoma A549 (ATCC^®^ CRM-CCL-185) cell line was cultivated in DMEM (Dulbecco’s Modified Eagle’s Medium, SIGMA^®^, San Luis, MO, USA) supplemented with 10% FBS (Fetal Bovine Serum, GIBCO^®^, Billings, MT, USA) and 1% penicillin–streptomycin (100 U/mL, Invitrogen^®^, Waltham, MA, USA). Cells were seeded in SPL™ T-75 flasks (2 × 10^4^ cells/cm^2^) and were incubated at 37 °C in a 5% CO_2_-humidified atmobead incubator. The medium was replenished every 2–3 days, and cells were detached from the flask through treatment with 0.25% trypsin–ethylenediaminetetraacetic acid (EDTA) (Thermo Fisher Scientific^®^, Waltham, MA, USA).

### 3.2. Three-Dimensional Alginate Scaffold Cell Culture

Sodium alginate was purchased from SIGMA^®^. Alginate-based 3D scaffolds were prepared using the following method: A solution at a concentration of 1.5% (*w*/*v*) was sterilized via autoclaving at 121 °C for 15 min. Then, 5 × 10^5^ A549 cells/mL were centrifuged for 5 min at 2000 rpm to form a pellet. After that, cells were homogenized into a 1.5% sodium alginate solution (1 mL). The resulting suspension was dripped into a 102 mM CaCl_2_ solution for encapsulation, forming beads [29]. The final concentration of cells was 40µL/bead = 4 × 10^4^/bead (25 beads/1 mL = 5 × 10^5^ cells), for each one. The 3D beads were cultivated in 24-well plates in the same culture conditions described above. Each bead had approximately 40 µL.

### 3.3. Characterization of 3D Alginate Scaffolds

#### Microscopy

Scaffolds with microencapsulated human lung cells were evaluated daily using an inverted phase-contrast optical microscope (Carl Zeiss^®^, Oberkochen, Germany) and a fluorescence microscope (In Cell Analyzer 2000, GE Healthcare^®^, Chicago, IL). The size and shape of the beads were recorded, and cell morphology and cell distribution within the scaffold were observed.

### 3.4. Viability and Density of A549 Cells inside 3D Alginate Scaffolds

#### 3.4.1. Viability Trypan Blue Assay

Alginate beads encapsulated with human lung A549 epithelial cells were monitored for 14 days to evaluate cell viability. Thereby, beads were removed from the 96-well (1 bead/well) plate at the stipulated times (24–368 h); then, ten beads were incubated in a water bath in a microtube at 37 °C for 40 min in 5 mL of sodium citrate solution (155 mM) for the dissolution of the hydrogel. After that, the recovered cells were centrifuged at 1500 rpm for 10 min at 22 °C. A cell pellet was obtained and suspended in 200 μL of culture medium and 0.4% trypan blue solution (Sigma^®^) (1:1, *v*/*v*). Viable cell counts were determined using a hemocytometer chamber. Three independent experiments were performed in sextuplicates.

#### 3.4.2. Viability Resazurin Assay

A resazurin assay was used as a second method to monitor cell viability and was performed at the same time as that described above. Cells were seeded in 96-well plates at different densities (2 × 10^4^; 1 × 10^5^; 5 × 10^5^ cells/well) for 24 h at 37 °C at a 5% CO_2_. Then, they were washed with phosphate-buffered saline (PBS), and 500 µM of resazurin solution (freshly prepared 1:10, *v*/*v*) was added per well (200 µL total volume). Incubation for 6 h at 37 °C in the dark followed [48]. Resorufin measurement was carried out using the excitation wavelength of 540 nm and emission of 590 nm in a microplate reader. Cells treated with 40% DMSO (*v*:*v*, diluted in culture medium) were used as death controls. Cells only treated with a culture medium were considered as live controls. The cell survival rate was calculated as follows: cell survival (%) = (OD resazurin)/(OD control) × 100% [48]. All samples were performed in duplicate. Each well contained a single microbead. Three independent experiments were performed.

### 3.5. Pb Culture

Pb (Pb18 strain) was originally isolated from a clinical case of paracoccidioidomycosis and preserved at the School of Pharmaceutical Sciences of Araraquara, São Paulo State University, Brazil. This strain was then inoculated in mice and transferred from the mouse lungs to semisolid Fava-Netto’s medium [49], with a pH value of 7.2 at 36 °C, and subcultured every 5–7 days to expand the yeast form. Yeast colonies obtained from the previous culture were recovered and subsequently cultivated in a liquid brain–heart infusion broth (BHI), containing 1% (*w*/*v*) glucose at 37 °C, in a shaker at 150 rpm for three days (exponential growth phase), followed by centrifugation and the adjustment of the inoculum based on a hemocytometer chamber.

### 3.6. Infection Assay of Pb

#### Two-Dimensional Monolayers of Lung Cells

A549 cells were adjusted to 2 × 10^4^ cells/well and cultivated in 24-well plates at 37 °C for 24 h. Aiming to increase the physiological significance of our models, a mix of ECM proteins containing fibronectin, fibrinogen, type I collagen, and type IV collagen was added in the sodium alginate solution before bead disposition. The concentration of each protein was 50 µg/mL. Then, 1 × 10^6^ yeast/mL were stained with 0.1% Calcofluor White (CFW) (Sigma^®^) for 30 min and subsequently deposited on wells in a ratio of 10:1 yeast/cell. This fluorophore binds fungi cell walls. The yeast and cells were further incubated for 24 h at 37 °C for interaction. After this period, the cells were washed twice with PBS (to remove yeasts that were not attached) and examined using fluorescence microscopy with excitation at 355 nm to prove fungi cells were present (GE Healthcare InCell Analyzer 2000). Naturally, monolayers with and without the ECM were studied.

### 3.7. Three-Dimensional Alginate Scaffolds

Differentially, 3D alginate scaffolds containing A549 lung cells were prepared as described previously, with some modifications. The ECM protein mix (fibronectin, fibrinogen, collagen I, and collagen IV, 50 µg/mL each) was added to the alginate solution and homogenized before the cells were added. The infection step was performed as detailed above and examined using the same equipment (GE InCell Analyzer 2000).

### 3.8. Confocal Microscopy

Confocal microscopy was performed (Carl Zeiss^®^ LSM 800) to verify the location of the yeasts in 3D alginate scaffolds. Prior to infection, yeasts were dyed with CFW 0.1% for 30 min, followed by the addition of fluorescein isothiocyanate (FITC) for 30 min. Plates were then washed with PBS to remove FITC excess. Images were analyzed in Zen Blue™ Software. Parameters such as sample thickness and depth, color crossing, and yeast position were analyzed.

### 3.9. Isolation of Pb from 3D Alginate Scaffold

Fungi cells were isolated from scaffolds using a dissolving solution (155 mM sodium citrate; 20 min in a heater at 37 °C). The number of viable fungi in infected 3D alginate scaffolds was determined by counting the colony-forming units (CFUs). This was carried out after the incubation period of 168 h (7 days) in the BHI Agar medium, and colonies were counted using the conventional method. This assay was performed in duplicate.

### 3.10. Statistical Analysis

The statistical analyses of the experiments described in viability, cell density, resazurin assay, and colony-forming units (CFUs) were performed using the two-way ANOVA with Bonferroni’s post-test, with multiple comparisons. All experiments were verified for the normality of the data. Statistical analyses were performed using the GradPadPrism 7 Software, and *p* < 0.001 was considered significant. All tests were performed in three independent experiments.

## 4. Results

### 4.1. Cell Seeding, Viability, Total Cell Counts, and Cell Metabolic Activity

The gelling process generated 3D alginate scaffolds that showed a spherical, bead-like format (Figure 1A) with an approximate diameter of 0.36 cm. Cell distribution was homogeneous within the scaffold, as seen by the nuclei stained with DAPI (Figure 1B). During the investigation, no empty areas were observed (cell absence). A549 cells cultivated as 2D monolayers exhibited cuboidal and polygonal morphologies (Figure 1C), attached to the culture flask, whereas their morphology shifted to a spherical form when cultivated in the 3D scaffold (Figure 1D). The limits of the SA bead inside the well were easily identified. The shadow aspect was due to the location of cells in many different planes.

Lung cell viability was assessed via Trypan blue staining after releasing and recovering the cells from the scaffolds. The viability of the encapsulated A549 cells, after a 5-day test period, ranged from 75% to 85%. Although most cells remained viable over time, a slight decrease in viability was observed for the cells seeded in the 3D scaffold (Figure 2A). Additionally, the total cell values were expressed as cell density. Overall, significant influence was found on cells’ growth in 3D scaffolds during 5 days when compared with the 2D culture (Figure 2B). It is essential to highlight that in the 3D model, there was an unequal distribution of nutrients, oxygen concentration, and pH gradients, caused by cellular cluster formation, which impacted cell proliferation and long-time viability maintenance.

Furthermore, the resazurin salt assay was performed to indicate the cell viability/metabolic activity for A549 cells seeded in the 3D alginate scaffold. Simultaneously, different cell densities were used on the colorimetric assay to decide the better one for the following assay, the infection model. When 2 × 10^4^ A549 cells were used for both culture models, it was possible to observe a more stable profile of viability based on metabolism, at an average of 95% after five days of culture for the 3D model, and 60% for the 2D culture (Figure 2C). The culture period was standardized for five days due to the log growth phase of the Pb18 strain, which was later employed to evaluate the host–pathogen interaction. Additionally, in the cell viability experiment based on cell metabolism detected using resazurin, it is possible to observe that the density of 2 × 10^4^ cells resulted in significant statistical differences. It was also evidenced that the cell metabolism suffered a decrease with time, which was not observed in the three-dimensional culture (Figure 2C, highlighted lines). Even though the Trypan blue analysis showed some decrease in the cell viability in the 3D model (Figure 2A), the cell metabolism remained stable and constant. When comparing other densities such as 1 × 10^5^ and 5 × 10^5^ between the monolayer and the 3D model, there were no significant differences, but it was also observed that the monolayer with 1 × 10^5^ presented a slight viability decline, ending the curve at around 76%. The same was not observed for the 3D model. Regarding the density of 5 × 10^5^, which is a large number of intentionally seeded cells, we could see that the monolayer 2D model also presented a viability of around 75% at the end of the experiment, different from the 3D model, which ended the curve with 95% of viability.

### 4.2. Infection Assay of Pb in 2D Monolayer and 3D-Cell-Seeded Alginate Scaffolds

Initially, we investigated the ability of the fungi to attach to 2D and 3D surfaces formed with A549 cells. A few yeast-like cells of the Pb18 strain were observed on the 2D A549 monolayers, while they were more abundant and internalized inside the 3D alginate scaffold containing these cells (Figure 3). In the presence of ECM proteins (Figure 4C,D), we saw an increased number of budding yeasts within the 3D alginate scaffold compared to when ECM components were not included (Figure 4A,B). However, yeasts with multiple buddings were noticed when the ECM proteins were added (Figure 4C,D).

### 4.3. Interaction of Pb with Cell-Seeded 3D Alginate Scaffolds

A confocal imaging technique was performed to verify the interaction between lung cells and Pb18 yeasts, which confirmed that yeasts-like cells (Figure 5A) were overlapping with A549 lung cells. The fungal cells labeled with CFW (blue) and the lung cells labeled with FITC (green) were observed in close contact, which suggests fungi adherence to the cells (Figure 5B,C). Different sections of the same analyzed field were represented, showing the presence of yeasts and A549 lung cells inside the 3D alginate scaffold in close contact (Figure 5D to Figure 5F).

### 4.4. Isolation of Pb from 3D Alginate Scaffold

The notable growth of Pb18 CFUs was observed after recovering them from the 3D alginate scaffolds and seeding them in a solid BHI Agar medium. During the 7-day incubation, it was possible to verify the increase in the CFUs (Figure 6). In the 2D culture, the presence of ECM components had no significant impact on the recovery of viable fungal cells after the infection assay. However, in the 3D alginate culture, the number of viable fungal cells recovered was significantly higher with the addition of ECM components.

## 5. Discussion

Fungal infections are highly relevant and impose a substantial burden on healthcare worldwide [50]. One of the most prevalent fungal diseases in Latin America is paracoccidioidomycosis, caused by a dimorphic fungus, *Paracoccidioides* spp. [51]. This fungus enters the host by the lungs where pneumocytes and pulmonary fibroblasts are present, embedded in components of ECM [5,6,44,45,46,52]. Various approaches have been developed to study the *Paracoccidioides*–host interaction [3,5,6,43,44,45,46]. However, there is a lack of microenvironmental data on the main entry point for many fungi, namely, the respiratory tract [4,5,6,52]. Researchers can now rely on in vitro study models ranging from simple 2D monolayer cultures to elegant and complex organ-on-a-chip platforms, which provide a better understanding of these interactions. Furthermore, it is also imperative that new biomarkers for these diseases, and their prognoses, be found. Finally, the identification of virulence factors is also of utmost importance [3,53,54,55,56], as the discovery of new therapies and physiological and immunological mechanisms remain unknown. According to the literature, we still do not have enough studies on the interaction of this fungus in 3D models. Therefore, we aimed to develop an alternative yet simple 3D model based on SA scaffolds to evaluate the host–pathogen interaction between human lung cells and *P*. *brasiliensis*. Three-dimensional in vitro models mimic the conditions found by the fungus in the living organism in the best possible way, with the exception of animal models.

Initially, we investigated the ability of the Pb18 fungus to attach to 2D and 3D surfaces seeded with A549 cells. One of the simplest, the human A549 adenocarcinoma cell line, has been used as an in vitro model for studying several diseases, including fungal infections [3,4,45,52,57]. Some positive attributes of this cell line are the feasibility of studying important processes such as adhesion [4,46,47,58,59,60], endocytosis, epithelial detachment, epithelial damage, and the presence of receptors that interact with the fungi cells [3,57,61].

The viability of A549 cells was determined via resazurin and trypan blue exclusion tests. Both methods showed the high viability of monolayer cells and cells encapsulated in alginate beads for five days of culture. Thus, these results reinforce the possibility of using this model in biomedical research [62].

Our study concluded that the 3D alginate scaffold did not generate excessive cytotoxicity in the encapsulated cells. Both 2D and 3D cultured cells presented high viability up until 48 h of cultivation. Only after 72 h of cultivation did a significant reduction in cell viability in the 3D model occur compared to the monolayer culture; however, it was still up to 80%. It is important to mention that most drug screening experiments using in vitro models generally last a maximum of 72 h [32,33,63], at which point, the viability measured in our experiments was within acceptable standards. Moreover, even at 96 h, the viability of the 3D scaffold was at 80%. An ideal 3D model should maintain high cell viability for the duration of the interaction experience, and this was achieved successfully using our 3D model. Literature studies show a similarity between the metabolic activity results for 3D cultures, where it gradually grows to reach its peak at 13 days, with a decrease after that [64].

One of our questions was how many cells should be seeded for these scaffolds. We tested different cell concentrations, and the 2 × 10^4^ cells/bead was the best one. When we performed the same experiment with more cells, there was a decrease at the end of the experiment, observed for both the 3D model and the monolayer model, which may have been due to the excessive number of cells present. In our experiments, starting at 2 × 10^4^ cells/bead, total cell counts revealed that each scaffold could yield approximately 4.3 × 10^5^ cells/bead after 72 h of cultivation, twice the initial seeded density. This result means that approximately three beads are necessary to obtain at least 1 × 10^6^ cells at the end of a 72 h experiment, a usual number used in cell cultures. These data are highly relevant for researchers who may want to use our proposed 3D model in drug screening/interaction studies.

Literature studies such as those by Wang [64] show a similarity between metabolic activity results for 3D culture, presenting the same graphic profile obtained in this experiment. By adding more cells (5 × 10^5^ cells/bead), one would expect, in principle, more cell–cell contact. However, the results revealed a metabolic activity level that was slightly below that obtained in the previous cell density (2 × 10^4^ cells/bead), starting at 48 h for the monolayer but being more present in a three-dimensional model. Another important factor in analyzing this curve is that due to the possible higher number of cells if longer cultivation times are used, a loss of viability is expected due to the excessive consumption of nutrients and the higher excretion of metabolites [65].

Regarding the morphological findings and analyzing several fields for cell distribution, we observed that the passage of light was hindered, generating opacity in the image. It was also perceived, comparing the images, that practically all the cells were translucent in 24 h, which points to high viability, and that approximately 50% of the cells became opaque after 144 h, which could be a result of increased internal granularity/cell death. Even though no direct assay for proliferation was employed, it was easy to notice a higher number of cells during the progression of the cell cultures, which points to cell proliferation.

Having established that the scaffolds maintain cell viability and probably allow cell proliferation, we turned to the characterization of the interaction between cells and the fungi. Our group has been studying this interaction for some time in monolayer cultures. We established a role for the ECM in a previous study [5]. Here, the effects of the ECM on fungi–cell interaction were more pronounced when those proteins were added to the culture medium. Moreover, an experiment in which the fungal inoculum was previously stained with CFW reinforces the importance of the ECM: several A 549 cells were found in close contact with the fungi, and this was upregulated in the experiment with the ECM. Given that this fluorescent dye acts on fungal wall structures, it differentiates yeasts from animal cells [66]. The effects of ECM proteins were intense for the A549 cell line, which was to be expected, given the nature of this cell (epithelial phenotype). These results corroborate the concept that A549 cells are excellent for Pb studies, as mentioned in previous references, such as the study that reinforced the importance of the Pb interactions with ECM proteins during the adhesion process [5,6,46,47,61]. Previous studies corroborate our findings, demonstrating the role of fibronectin and fibrinogen glycoproteins during *P. brasiliensis* conidia adherence to A549 cells. For example, previous authors demonstrated that the fungal adherence process in lung cells was facilitated by both proteins [67]. Several other studies described the interaction of these molecules with surface proteins of *P. brasiliensis* [5,6,44,45,46,47].

In our model, the proximity between the host cells and fungal cells within the 3D framework was observed, which might be related to adhesion molecules of fungal cells and ligands on target cells, as was described before in a 2D model [4,52,60,67,68,69,70]. Furthermore, given that the fungi cells were not added during the gelation process but after it, the only way to reach the inner layers of the 3D scaffold, where the host cells were located, would have been by migration. Therefore, one can safely conclude that the 3D alginate model proposed here can allow cell and fungi migration, corroborating its potential as a 3D infection model [68,69].

Another study that corroborates our findings has demonstrated that the survival of *Mycobacterium tuberculosis* into 3D collagen–alginate microbeads was much higher than in 2D cultures [71]. This report also observed that multinucleated giant cells’ formation, a typical feature of human tuberculosis, was seen in the scaffolds, similarly to those that occur in human patients with this disease, as well as the upregulation of protease activity at the infection site, a common event in tuberculosis pathogenesis [72].

Moreover, the combination of 3D alginate microbeads with type I collagen significantly reduced immune cell death after infection, demonstrating that this ECM constituent improves the host control of mycobacteriosis [71]. Eventually, T cells responsive to specific antigens proliferated in infected microbeads and secreted cytokines that play a crucial role in the host immune response to the bacterium [73,74]. Thus, their system proved to be highly flexible to incorporate both matrix components and primary cells within alginate beads, making it feasible to study biological/immunological mechanisms that occur during the infection of peripheral blood mononuclear cells [17,71].

Another interesting verification in the images is that in the 3D models, both cells and yeasts were found in the folds created during the alginate gelation process. These folds are the outer-most region of the scaffold, and the presence of yeasts suggests these are the entry point for the fungi [31].

Despite the fact that, with this investigation, we were able to observe interesting results, such as the yeast being able to migrate the interior of the gel, according to image sections, and that this model is biocompatible with A549 cells, our study has some limitations. For example, there is a need for more advanced microscopy studies that can respond assertively to the location of the fungus, in addition to histological techniques that, in addition to elucidating the presence of yeast in tissues, would help demonstrate where the fungus is located. It is also important to note that systematic tests should be conducted in future studies to demonstrate the superiority of the three-dimensional model to the monolayer culture, in order to confirm the preliminary results presented.

In conclusion, we have demonstrated that a 3D alginate bead scaffold seeded with A549 cells is a promising model for *Paracoccidioides* interaction/infection studies. In addition, our model allowed the easy addition of important components, such as those ECM molecules present in the lungs. It is also evident that the fungal cells interacted with the scaffolds, penetrated them, and reached the lung cells that were encapsulated inside. This creates several possibilities for our proposed 3D model in biomedical research. The obvious example would be testing an anti-fungal drug using the 3D model and then plating the retrieved contents. It is worth mentioning that the advent of organ-on-a-chip platforms is still in its infancy, with several points left to address, such as the scale of the constructs (when compared to real organs), the fact that the experience gathered so far has been directed to drug screening, and not to host–pathogen interactions, and obviously, their extremely high cost [3,43]. Therefore, more straightforward but reliable models, such as the one reported here, might produce important data while the organ-on-a-chip models evolve. What remains to be analyzed is which pathogenicity mechanisms/molecules, such as adhesins and invasins, the fungal cells use during interaction on this 3D scaffold.

## Figures and Tables

**Figure 1 jof-09-00634-f001:**
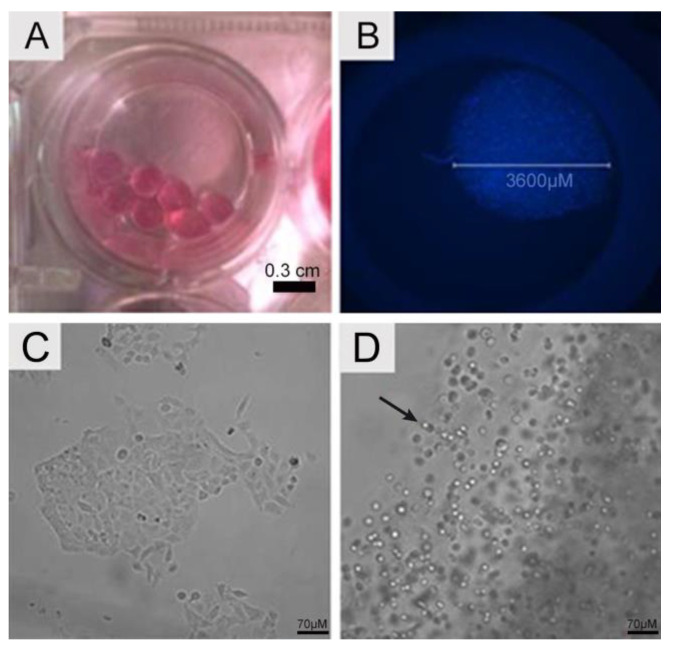
Scaffold morphology. (**A**) Macroscopic image of alginate scaffolds recorded with a camera. (**B**) Fluorescence microscopy showing a single alginate scaffold in a 96-well plate. The beads exhibited 3600 µm and contained A549 encapsulated cells, homogeneously distributed within the beads. The nuclei of cells were stained using DAPI (blue). Cellular morphology. (**C**) A549 cells in 2D monolayer culture displaying cuboidal and polygonal morphology. (**D**) A549 cells in 3D alginate scaffolds. Note that the morphology is now rounded (arrow). The micrographs were recorded using bright field fluorescence microscopy.

**Figure 2 jof-09-00634-f002:**
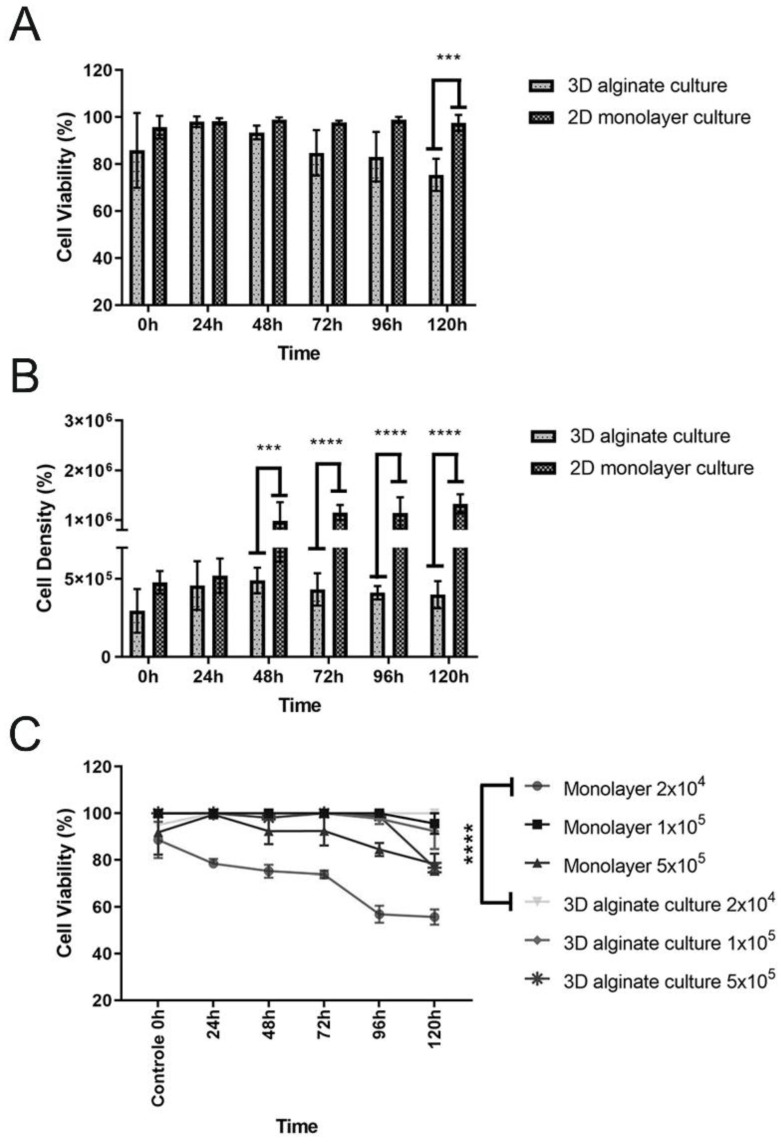
(**A**) Cellular viability measured during 5 days (0–120 h). Live lung cells inside the 3D scaffolds remained at an average of 75% after 5 days of cultivation, while the viability at 2D culture was 100%. Values obtained via the mean ± SD are the results of a two-way ANOVA. *** *p* = 0.0009 3D versus 2D culture (120 h). (**B**) Total cell counts measured during 5 days (0–120h). Cell growth inside the 3D scaffolds demonstrate a significant variation when compared to 2D culture. Values obtained via the mean ± SD are the results of an two-way ANOVA. **** *p* < 0.001 (3D versus 2D culture) (48 h; 72 h; 96 and 120 h). (**C**) Metabolic activity of A549 cells. Assays for 2D and 3D seeded cells in alginate scaffold were performed using the resazurin salt assay. Among the tested cell densities, it was possible to observe all curves starting from 100% viability (control/0 h). The curves followed distinct patterns of cell viability based on the ability to convert resazurin to resorufin. The reading was performed at 590 nm.

**Figure 3 jof-09-00634-f003:**
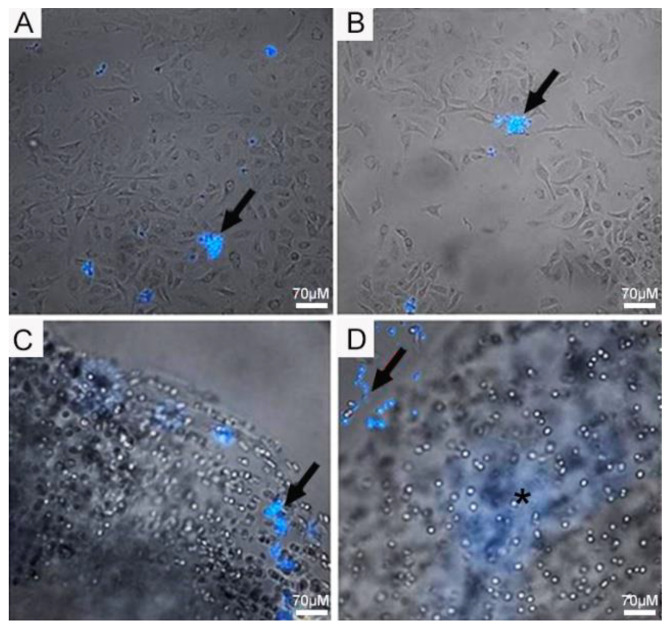
Infection assay of Pb18 fungi in A549 cells cultivated in 2D monolayers and 3D alginate scaffolds. (**A**,**B**) A549 cells in 2D monolayers infected with Pb18 yeast cells. (**C**,**D**) A549 cells seeded in 3D alginate scaffolds infected with Pb18 yeast cells. Yeast cells (arrows) were previously stained in blue with CFW and were more abundant in the 3D culture. The culture medium was supplemented with a mix of ECM proteins. Note the blurred area (*); this is the internalized fungi on the scaffold, which, being tridimensional, creates this blurred area in blue, due to the staining with CFW.

**Figure 4 jof-09-00634-f004:**
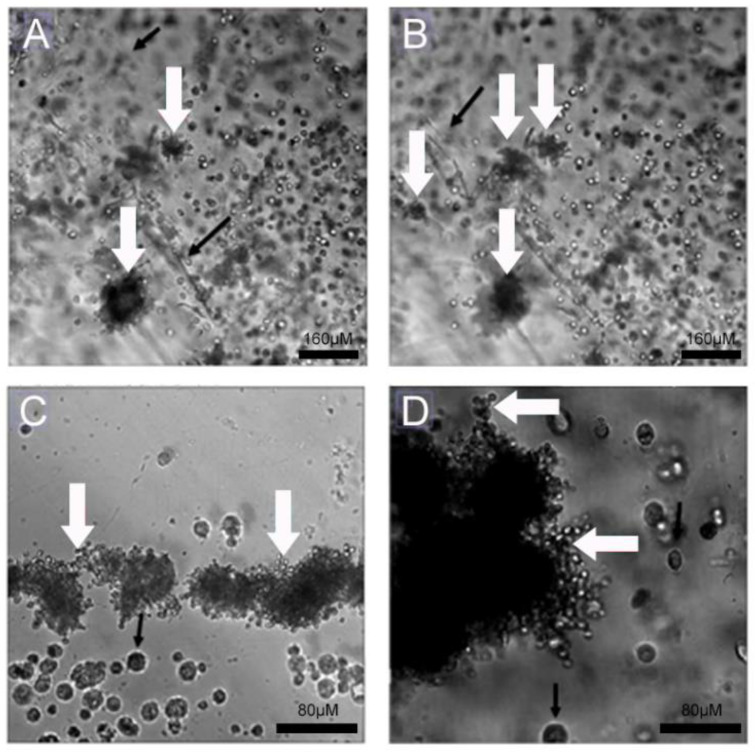
Infection assay of Pb18 fungi in A549 cells cultivated in 3D alginate scaffolds with and without ECM proteins. (**A**,**B**) A549 cells seeded in 3D alginate scaffold, infected with Pb18 strain, without addition of ECM proteins. (**C**,**D**) A549 cells seeded in 3D alginate scaffold, infected with Pb18 strain, in the presence of ECM proteins. Budding yeasts, typical of daughter cells generated from only mother cells, can be seen (white arrows); some are in close contact to A549 cells (black arrows).

**Figure 5 jof-09-00634-f005:**
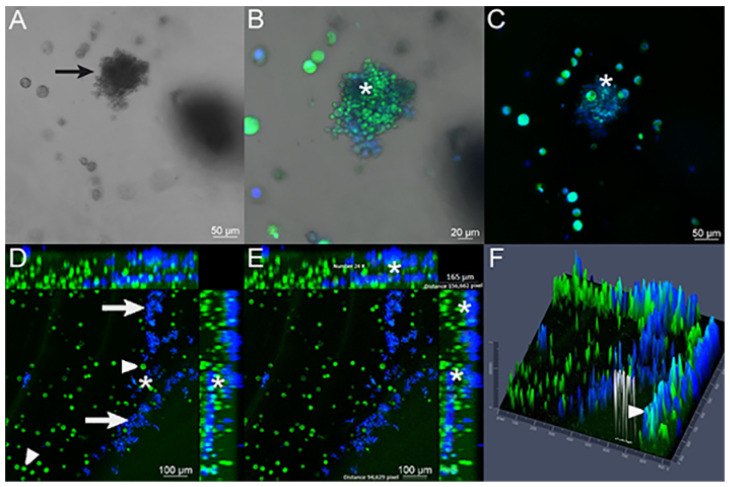
Infection assay verified with confocal microscopy of 3D alginate scaffold infected with the Pb18 strain. (**A**) Bright field image showing budding yeasts (arrow) in association with A549 lung cells. (**B**,**C**) Merged images of FITC-labeled A549 cells (green) and CFW-labeled Pb18 yeasts (blue) inside the 3D alginate scaffold. It is possible to observe the presence of yeast cells in close contact to A549 cells (*). (**D**) A549 cells (arrowhead) and yeasts (arrow). There is an elevated number of yeasts inside the scaffold, and it is possible to confirm the proximity between A549 host cells and fungal cells (*). (**E**) Presence of the overlap between blue and green colors (*), showing the contact between the yeasts and lung cells, analyzed in 24 points. The analyzed field thickness was 165 μm, obtained using the ZenBlue software. (**F**) A 2.5-dimensional image generated using the Zen Blue software. This result demonstrates a graphical representation (this time using height and depth) of the previous images, showing different height peaks for the colors green and blue, individually. The height measurements of the colors can elucidate where the cells and yeasts are located (an upper or lower section). The arrowhead shows the combination of green and blue colors, in the same height/depth, resulting from the proximity between the yeasts (that were plated outside the three-dimensional model) and the cells that were plated inside the 3D model.

**Figure 6 jof-09-00634-f006:**
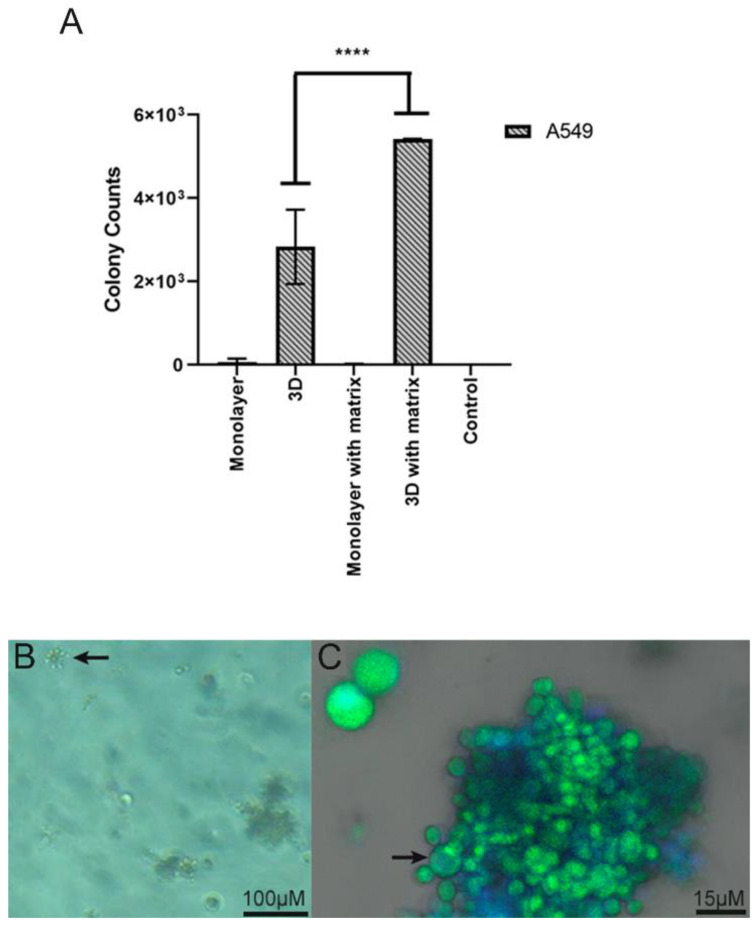
(**A**) Number of yeasts isolated by CFUs derived from the 3D alginate scaffold in association with lung cells, and the number of colonies represents yeasts cells that are capable of surviving the interaction and replicating. A semi-quantitative assay was performed, in duplicate, in which the number of CFUs was recorded in each culture condition: monolayer, 3D (three-dimensional alginate sodium bead), monolayer with matrix (monolayer with addition of ECM proteins), and 3D with matrix (three-dimensional alginate sodium bead with addition of ECM proteins). Values given by the mean ± SD are the results of a two-way ANOVA. **** *p* = <0.0001. (**B**) Yeast cells derived from the 3D alginate scaffold. Bright field image showing budding yeasts (arrow). (**C**) Pb18 yeasts presenting classical morphology. Mother cell (arrow) and attached daughter cells.

## Data Availability

The data presented in this study are available on request from the corresponding author.

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
