# Peer review of "Alginate-Based 3D A549 Cell Culture Model to Study *Paracoccidioides* Infection"

_jof, 2023, doi:10.3390/jof9060634_

Round 1
Reviewer 1 Report
In this paper, the authors used an alginate-bead model with a lung epithelia cell line to infect them with Paracoccidiodes brasiliensis. The authors claim that such 3D models could benefit the future research into Pb pathogenesis.
Overall, I am not convinced by the data presented here.
- In the introduction, the authors extort the many virtues of 3D models, from mechanical stresses to specific metabolism. However, none of those are tested/measured here. Actually, the human cells round up and die quicker in the alginate, which in itself is not too promising. (Although that is confusing, as figure 2A and 2D seem to contradict each other, if the axis label "cell vaibility" is correct).
- The finding that Pb is in association with the lung cells is not convincing, based on essentially a single (or few) proximity event in an otherwise not too clear micrograph. There is no quantification, or at least an attempt to e.g. re-isolate the cells from the 3D model to show the association. The use of the 2.5D image is not explained well, and seems not too convincing to show what the authors claim in the text
- The increased cell numbers of Pb are similarly unconvincing, at least a test for growth in absence of the human cells would be required (and even then, what is the conclusion, what is a potential reason for that?).
Overall, I do not see what the findings of this manuscript are. The model itself does not seem to be an improvement over other models (or at least it has not been shown here). Why would an embedded-in-alginate 3D model be better as a lung infection model, a type of infection where the fungus meets an essentially 2D epithilal cell layer first? Thus I do not really agree with the conclusion that this is an excellent model for the study of the interaction.
Overall, I am sorry to say that therefore I am not convinced by the design of the model, and the data is not conclusive enough to support the statements in the discussion. A more thorough investigation (with some quantitative data) may prove that there are applications for this model, which I cannot see yet.
Author Response
Reviewer: In the introduction, the authors extort the many virtues of 3D models, from mechanical stresses to specific metabolism. However, none of those are tested/measured here. Actually, the human cells round up and die quicker in the alginate, which in itself is not too promising. (Although that is confusing, as figure 2A and 2D seem to contradict each other, if the axis label "cell vaibility" is correct).
Answer: The "introduction" section is intended to present data about the work that will be presented. That is why we mentioned the advantages of the 3D cell cultures. Obviously, not every aspect mentioned in the "introduction" has to be measured in the paper. The present work aims to investigate the use of a natural polymer to study fungal interactions. It was a preliminary study, to obtain something that is more promising new investigations advancing what was done is necessary.
The figure 2 goes A-C, there's no 2D figure! Also, these figures don't contradict each other because we can see that viability on figure C remains in 80% for almost all measurements. Figure 2A shows that viability is 80% until 96h. The following experiments were conducted in a gap of 0-24h of interaction, therefore there's no motive that the cells inside the 3D model have to remain viable for so long, but regardless we have shown on our work that this model can maintain the cells alive for a longer period if we wanted. Moreover, the 2A viability measurement was based on a manual method (trypan blue), and figure 2C was made after an automatic method was performed (that is based on the metabolic activity by the cells). Because trypan blue is a manual exclusion method, we could presume that the results could be a little bit different because the principles are different.
Reviewer: The finding that Pb is in association with the lung cells is not convincing, based on essentially a single (or few) proximity event in an otherwise not too clear micrograph. There is no quantification, or at least an attempt to e.g. re-isolate the cells from the 3D model to show the association. The use of the 2.5D image is not explained well, and seems not too convincing to show what the authors claim in the text.
Answer: The proximity events were abundant, and clearly demonstrated in Figure 5. Given that the fungal cells were added AFTER the human cells were inserted inside the scaffolds, these fungi cells could only be seen inside the scaffold, near the human cells, trough migration. The figure you mentioned was added to show that the interaction (blue and green proximity) is three-dimensional. You can also observe that in figure 2F there is a change in color from green to blue with a "bluish" green tone.
Reviewer: The increased cell numbers of Pb are similarly unconvincing, at least a test for growth in absence of the human cells would be required (and even then, what is the conclusion, what is a potential reason for that?).
Answer: There is no need for this experiment, because human cells will not grow properly in fungi culture medium.
Reviewer: Overall, I do not see what the findings of this manuscript are. The model itself does not seem to be an improvement over other models (or at least it has not been shown here). Why would an embedded-in-alginate 3D model be better as a lung infection model, a type of infection where the fungus meets an essentially 2D epithilal cell layer first? Thus I do not really agree with the conclusion that this is an excellent model for the study of the interaction.
Answer: Indeed, the first contact of the fungal cells with the human cells, in the lung, is with a 2D epithelium. However, the fungal cells dont stay there, they will break through and invade the connective tissue bellow the ep. Therefore, the environment is 3D. Our model recapitulate this, and also recapitulate the epithelium-stroma interaction, given that we added ECM proteins in the scaffold. We can see clearly how this is an improvement over a simple 2D monolayer.
Reviewer: Overall, I am sorry to say that therefore I am not convinced by the design of the model, and the data is not conclusive enough to support the statements in the discussion. A more thorough investigation (with some quantitative data) may prove that there are applications for this model, which I cannot see yet.
Answer: We disagree, and the other two reviewers also would disagree, given the positive comments they made.
Reviewer 2 Report
The authors present their research on the construction of a 3D model of alginate with lung adenocarcinoma cells to be infected with paracoccidiodis brasiliensis. This is an interesting and overdue study as the 3D model comes much closer to the real life experience than 2D plates. Paracoccidiodomycosis is of tremendous importance in South America as health problem and therapeutic challenge.
The English requires improvement.
Lines 116, 176: "2. D cell culture" Should it not be "2D cell culture"?
Lines 125, 187. "3. D alginate scaffold cell culture" Should it not be "3D alginate scaffold cell culture"?
Line 168: "innoculated" omit one n
Line 185: "examined using fluorescence microscopy with excitation at 355 nm" This waveline is UV and cytotoxic.
Line 345:"cell-cel1l contact" is it "cell-cell contact"?
Author Response
Reviewer: The authors present their research on the construction of a 3D model of alginate with lung adenocarcinoma cells to be infected with paracoccidiodis brasiliensis. This is an interesting and overdue study as the 3D model comes much closer to the real life experience than 2D plates. Paracoccidiodomycosis is of tremendous importance in South America as health problem and therapeutic challenge.
Reply: We thank you for your positive review.
Reviewer: The English requires improvement. Lines 116, 176: "2. D cell culture" Should it not be "2D cell culture"? Lines 125, 187. "3. D alginate scaffold cell culture" Should it not be "3D alginate scaffold cell culture"? Line 168: "innoculated" omit one n Line 185: "examined using fluorescence microscopy with excitation at 355 nm" This waveline is UV and cytotoxic. Line 345:"cell-cel1l contact" is it "cell-cell contact"?
Reply: The paper was, once again, reviewed. All the sentences that were cited were corrected (highlighted in yellow). Regarding the 355 nm waveline, we clarify that this was not a viability assay. Calcofluor was used to bind to fungi cell walls and, therefore, mark them in blue, after the mentioned WV was used. Excitation in this WV, for calcofluor, is 100%. Therefore, it was not a problem that the WV is citotoxic. We added a sentence explaining this (highlighted in yellow).
Reviewer 3 Report
This paper carefully and clearly describes the effect on viability of cultured human cancer cells when yeast cells of Paracoccioides brasiliensis are added, comparing the conditions when cancer cells are grown in a monolayer as clumps in a sodium alginate gel. The authors found that at a ratio of 10 yeasts per cancer cell, the cancer cells lost little viability over 72 hours. Adding fibrinogen glycoproteins and fibronectin increased the growth of fungal cells and their contact with cancer cells. The authors propose the clumped cell model as a better approximation of fungal effects on mammalian cells than tissue monolayers. Although a proposed use of this model is to study antifungal drugs in a more biologic setting than culture medium, that would require a study of the effects of the model on the fungal cell, not the mammalian cell. The results here are relevant to studies of fungi on mammalian cell biology, not mammalian cell effects on fungi.
A priori, one would expect that the effect of viability of cancer cells would depend on cell number, density and metabolic activity of cancer and fungal cells, nutrition and oxygen from the culture medium and probably contiguity between fungal and cancer cells. There are obvious limitations to measuring these parameters in this model and it would be preferable for the authors to point them out clearly. One issue is how to quantitate how many fungal cells are inside the clump, attached to the clump or floating around the clump. Another is measuring the metabolic activity of slow growing fungal cells on which many new budding cells remain attached. A third is how to quantitate the contiguity of cancer cells and fungal cells within a clump of 20,000 cancer cells. The photomicrographs provided give some qualitative data but the photos are of low magnification and in some, difficult to see (see Figure 1B). Abbreviations of “SA” (line 126) and “OOC” (line 411) need explanation. The bottom of the vertical axis in figures 2B and 6A are incorrectly labeled as zero.
Author Response
Reviewer: This paper carefully and clearly describes the effect on viability of cultured human cancer cells when yeast cells of Paracoccioides brasiliensis are added, comparing the conditions when cancer cells are grown in a monolayer as clumps in a sodium alginate gel. The authors found that at a ratio of 10 yeasts per cancer cell, the cancer cells lost little viability over 72 hours. Adding fibrinogen glycoproteins and fibronectin increased the growth of fungal cells and their contact with cancer cells. The authors propose the clumped cell model as a better approximation of fungal effects on mammalian cells than tissue monolayers. Although a proposed use of this model is to study antifungal drugs in a more biologic setting than culture medium, that would require a study of the effects of the model on the fungal cell, not the mammalian cell. The results here are relevant to studies of fungi on mammalian cell biology, not mammalian cell effects on fungi.
Answer: We agree with the reviewer, and appreciate the positive comments. The paper was evaluated again for English mistakes (highlighted in yellow).
Reviewer: One issue is how to quantitate how many fungal cells are inside the clump, attached to the clump or floating around the clump. Another is measuring the metabolic activity of slow growing fungal cells on which many new budding cells remain attached. A third is how to quantitate the contiguity of cancer cells and fungal cells within a clump of 20,000 cancer cells.
Answer: Regarding the first issue, we used calcofluor as a "specific" fluorophore, given that it has affinity to the fungi cell walls. Therefore, everything marked in blue after reading in 355 nm is fungi cells. Regarding the second issue, another study would be necessary to properly evaluate the fungi metabolism. And, regarding the third commentary, we used the contiguity of human cells (marked in green) with fungi cells (marked in blue) as an initial evaluation. More advanced techniques can further elucidate this.
Reviewer: The photomicrographs provided give some qualitative data but the photos are of low magnification and in some, difficult to see (see Figure 1B). Abbreviations of “SA” (line 126) and “OOC” (line 411) need explanation. The bottom of the vertical axis in figures 2B and 6A are incorrectly labeled as zero.
Answer: Figure 1B clearly shows the alginate bead scaffold with A549 encapsulated cells, homogeneously distributed within the beads. It is possible to see each rounded cell marked in blue. We dont understand why the reviwer thinks it has low quality. The SA and OOC abbreviations were removed. The "zero" labeling in FIG2B and FIG6A are correct (zero cells and zero colonies, respectively).